# Strategic Doping for Precise Structural Control and Intense Photocurrents Under Visible Light in Ba_2_M_0.4_Bi_1.6_O_6_ (M = La, Ce, Pr, Pb, Y) Double Perovskites

**DOI:** 10.3390/nano15131039

**Published:** 2025-07-04

**Authors:** Tirong Guo, Wen Tian Fu, Huub J. M. de Groot

**Affiliations:** Leiden Institute of Chemistry, Leiden University, Einsteinweg 55, 2300 RA Leiden, The Netherlands; t.guo@lic.leidenuniv.nl (T.G.); w.fu@chem.leidenuniv.nl (W.T.F.)

**Keywords:** double perovskite, doping, space group, electronic band structure, efficient photovoltaic materials

## Abstract

Developing functional perovskites is important for advancing solar energy conversion technologies. This study investigates the effects of dopants on the structural, optical, electronic, and solar conversion performances of Ba_2_M_0.4_Bi_1.6_O_6_ double perovskites. X-ray diffraction (XRD) and Rietveld refinement confirm crystallization in the *I*2*/m* space group (M = La, Ce, Pr, Pb), and *Fm*
$\bar{3}$*m* and *I*2*/m* space groups (M = Y). The B1-O-B2 structure modulates to highly ordered (M = La, Y), partially ordered (M = Pr), or disordered (M = Ce, Pb). UV-vis spectra show strong light absorption, with Tauc plots estimating ~1.57 eV (M = La) and ~1.73 eV (M = Pr) optical band gaps. Under AM 1.5G illumination, the M = La photoelectrode generates photocurrents of 1 mA cm^−2^ at 0.3 V_RHE_, surpassing M = Ce and Pb (1 μm, 4-times spin-coating). Increasing its thickness to 7.7 μm (4-times dip-coating) further enhances the photocurrents to 2.3 mA cm^−2^ at 0.2 V_RHE_, outperforming all counterparts due to improved stability. Fine-tuning crystal and electronic structures via strategic B-site doping provides a new route for engineering Ba_2_Bi_2_O_6_-based double perovskites for broad solar energy conversion applications.

## 1. Introduction

Perovskites have emerged as promising photovoltaic materials since first reported by Kojima et al. in 2009, attracting attention for their excellent light-absorbing properties [1]. Their low cost, high performance, and structural flexibility have positioned them at the forefront of solar energy harvesting–conversion technologies. Among them, inorganic oxide perovskites stand out for their superior thermal and chemical stabilities compared to their organic counterparts [2,3,4]. Recent advances in layered perovskites (e.g., Bi_2_WO_6_) have also highlighted their promising photoelectrochemical performance [5,6]. Theoretical studies further reveal that structural, electronic, and optical properties critically impact the optoelectronic characteristics of perovskite oxides [4,7]. With their adjustable band gaps, compositional versatility, and favorable structural defects, these materials show great promise for renewable energy applications [4,7,8]. However, the experimental understanding of the structure–composition–property interplay remains limited, necessitating systematic investigation to unlock their full potential.

The prototype perovskite oxide semiconductor Ba_2_Bi_2_O_6_ (BBO), first synthesized in the early 1960s, has since attracted considerable research interest [9,10,11]. Investigations into the crystal and electronic structure of the semiconducting BBO compound unveiled an ordered double-perovskite structure with mixed valence states of Bi^3+^ and Bi^5+^ [12]. The oxygen-bond-stretching displacements in the BBO material cause a charge density wave (CDW) distortion and the alternating arrangement of two inequivalent oxygen-surrounded Bi octahedrons, leading to the charge disproportionation into Bi^3+^ and Bi^5+^ and doubling of the unit cell [11,13,14]. To emphasize the double-perovskite oxide structure, the chemical formula of the BBO compound is often written as Ba_2_Bi_2_O_6_, rather than the simpler perovskite form, BaBiO_3_, following the general formula A_2_B_(1)_B_(2)_O_6_, where Ba^2+^ ions occupy the A-sites, Bi^3+^ ions occupy the B1-sites, and Bi^5+^ ions occupy the B2-sites.

Unlike simple perovskites, such as SrTiO_3_, BaTiO_3_, and LaFeO_3_, the BBO double-perovskite framework offers considerable compositional and structural flexibility [7]. By doping external ions at the A-site cations, B1- or B2-site cations, or oxygen anions, the electronic band structure and corresponding electronic properties can be effectively modulated [11,14,15,16,17]. In addition, the BBO material has high photocatalytic activity under visible light [18]. This activity has been further enhanced by systematically doping rare-earth elements into the BBO parental system, with the degree of improvement significantly depending on the choice of dopant ions [19,20]. Previous studies on modifying the BBO compound by doping pentavalent Nb^5+^ ions at the B2-sites have led to p-type semiconducting thin-film photoelectrodes for efficient photoelectrochemical (PEC) proton reduction [21] and n-type semiconducting thin-film photoelectrode systems with oxygen vacancies for high-photocurrent PEC water oxidation [22,23]. Tuning the electronic band structure with external ions is important for achieving high PEC performance [22,24,25].

To explore the engineering possibilities of the chemical composition, the crystal structure, and the electronic band structure of the BBO parent compound, we perform partial B-site doping with various trivalent and tetravalent ions for preparing a series of Ba_2_M_0.4_Bi_1.6_O_6_ (M = La, Ce, Pr, Pb, Y) double perovskites based on our previous concentration screening study [26]. X-ray diffraction (XRD) results and Rietveld refinement results reveal the formation of double-perovskite Ba_2_M_0.4_Bi_1.6_O_6_ with M = La, Ce, Pr and Pb in the monoclinic *I*2*/m* space group, while the Ba_2_M_0.4_Bi_1.6_O_6_ with M = Y shows two double-perovskite phases, with one in the cubic *Fm*
$\bar{3}$*m* space group and the other in the monoclinic *I*2*/m* space group. The lattice parameters exhibit a contraction tendency due to the reduction in the ionic sizes of the dopant ions and the shrinkage of the oxygen octahedra. In Ba_2_M_0.4_Bi_1.6_O_6_, doping with M = La and Y promotes a highly ordered B1-O-B2 arrangement, whereas doping with M = Ce and Pb introduces disorder at the B-sites. Meanwhile, doping with M = Pr results in partial ordering of the B-sites. UV-vis absorption spectra reveal that all samples have excellent light-absorption properties across a broad spectrum from 300 nm to 1000 nm. Double perovskites doped with M = La and M = Pr display optical band gaps, *E*_g_, of approximately 1.57 eV and 1.73 eV, respectively. Under one sun’s illumination, the photoelectrodes Ba_2_M_0.4_Bi_1.6_O_6_ with M = La achieved a high photocurrent density of 1 mA cm^−2^ at 0.3 V_RHE_ for the 1 μm thin-film spin-coated sample, and a further enhanced photocurrent density of 2.3 mA cm^−2^ at 0.2 V_RHE_ was obtained for the 7.7 μm thicker dip-coated film, surpassing all other competing samples. Thin-film overlayer protection and heterojunction construction could be promising strategies to further advance these materials for efficient optoelectronic, photovoltaic, and solar fuel applications [23,27].

## 2. Materials and Methods

### 2.1. Preparation of Powder Samples

Stoichiometric amounts of starting reagents BaCO_3_ (99.95%, Alfa Aesar, Haverhill, MA, USA), Bi_2_O_3_ (99.9%, Chempur, Mumbai, India), La(NO_3_)_3_·6H_2_O (99.9%, Alfa Aesar), Ce(NO_3_)_3_·6H_2_O (99%, Sigma Aldrich, St. Louis, MO, USA), Pr(CH_3_CO_2_)_3_·1.4H_2_O (99.9%, Thermos Scientific, Waltham, MA, USA), and Pb(CH_3_CO_2_)_2_·3H_2_O (99.5–102%, Honeywell, Charlotte, NC, USA) were calculated and weighed according to the chemical compositions of Ba_2_M_0.4_Bi_1.6_O_6_ (M = La, Ce, Pr, Pb); they were then mixed and ground in an agate mortar for at least 30 min, with 20 drops of ethanol. The finely ground reagents were transferred to an alumina crucible and annealed in a muffle furnace at 950 °C for 12 h to form double perovskites by solid-state reactions. The samples were naturally cooled to room temperature in the furnace and were subsequently ground for 30 min to obtain fine powders of Ba_2_M_0.4_Bi_1.6_O_6_ (M = La, Ce, Pr, Pb). For the preparation of the Ba_2_Y_0.4_Bi_1.6_O_6_ sample, the BaCO_3_ and Bi_2_O_3_ reagents were sourced from the same suppliers as mentioned above, while two different salts of yttrium, Y_2_O_3_ (99.99%, Acros Organics, Geel, Belgium) and Y(CH_3_CO_2_)_3_·4H_2_O (99.99%, Thermos Scientific), were used to compare the crystal diffraction patterns of the resulting compounds, adopting the same solid-state reaction process as described above.

### 2.2. Preparation of Film Photoelectrodes

To benchmark the solar-to-current conversion performance, Ba_2_M_0.4_Bi_1.6_O_6_ (M = La, Ce, Pr, Pb) double perovskites were deposited as highly crystalline thin-film photoelectrodes. First, 0.5 M bismuth-containing solution was prepared by dissolving 1 mmol Bi(CH_3_CO_2_)_3_ (99.99%, Alfa Aesar) in 2 mL acetic acid solution (glacial acetic acid and Milli-Q water in a 3:1 volume ratio). Then, 1.25 mmol Ba(CH_3_CO_2_)_2_ (99.0–102.0%, Supelco, Bellefonte, PA, USA) was added to the abovementioned bismuth solution with magnetic stirring until full dissolution. According to the composition of Ba_2_M_0.4_Bi_1.6_O_6_ (M = La, Ce, Pr, Y, Pb), 0.25 mmol of La(CH_3_CO_2_)_3_·1.5H_2_O (99.99%, Alfa Aesar), Ce(CH_3_CO_2_)_3_·1.5H_2_O (99.99%, Alfa Aesar), Pr(CH_3_CO_2_)_3_·1.4H_2_O (99.9%, Thermos Scientific), Y(CH_3_CO_2_)_3_·4H_2_O (99.99%, Thermos Scientific), and Pb(CH_3_CO_2_)_2_·3H_2_O (99.5–102%, Honeywell), respectively, was added to the solution and continuously stirred magnetically overnight until fully dissolved at room temperature. Subsequently, 0.75 mL 2-methoxyethanol (anhydrous, 99.8%, Sigma Aldrich) was slowly added to the abovementioned solution to adjust the viscosity of the sol–gel precursor solution.

Au substrate electrodes of 0.25 mm thickness and 25 × 25 mm initial dimensions were purchased (≥99.9975%, Thermo Scientific Chemicals) and cut to dimensions of 1.25 × 2.5 cm^2^. Prior to the coating of any precursor solutions, Au substrates were polished with the sanding paper (Matador waterproof P1000, Starcke, Melle, Germany) and then cleaned with acetone and Milli-Q water in an ultrasonic bath for 15 min each. The cleaned Au substrates were carefully dried using compressed air and stored in pre-cleaned glass vials for further use. Thin-film photoelectrodes processed via the spin-coating method followed a preset program: spin-coating 100 μL precursor solution on the Au substrate at 2500 rpm for 30 s, followed by a higher spin rate of 3500 rpm for another 30 s. After the spin-coating process, the substrate was dried on a hot plate at 80 °C for 1 min and cooled down to room temperature. The sample was then placed in an alumina crucible and introduced into a furnace at a temperature of 550 °C. After allowing it to settle in the furnace chamber, the temperature was manually raised to 950 °C. The sample was maintained at this temperature for 1 h to facilitate crystallization. It was removed and exposed to ambient air when the furnace had cooled down naturally below 700 °C. The coating–calcination process was repeated 4 times. In the last cycle, the sample was kept in the furnace until it had cooled down naturally to room temperature. The thin-film spin-coated undoped parental BBO photoelectrodes were also prepared from the precursor solution containing the barium and bismuth elements at a 1:1 ratio for comparison.

Thicker film photoelectrodes were processed via the dip-coating method, adopting a similar coating–calcination cycle protocol, as described above. Care must be taken to avoid excess flowable precursor sol–gel solution on the Au substrate, since it may lead to local gel aggregation and cause inhomogeneities during solvents volatilization. After the dip-coating process, the Au substrates were dried in a convection oven at 80 °C for 2 min before being transferred to the muffle furnace for calcination. The coating–calcination process was repeated for 4 cycles, with the samples remaining in the furnace until it cooled to room temperature in the final cycle.

### 2.3. Material Characterization

The crystal structures of sample powders and film photoelectrodes were characterized by X-ray diffraction (XRD) (PANalytical SR 5161). The Rietveld method was applied to refine the crystal structures and lattice parameters of the powdered samples [28,29]. The UV-vis optical absorption properties were measured with a UV-vis spectrometer (Agilent, Cary 60) from a uniform suspension of 1 mg mL^−1^ Ba_2_M_0.4_Bi_1.6_O_6_ (M = La, Ce, Pr, Pb, Y) fine powder in 96% ethanol, respectively [30]. The surface morphologies of Ba_2_M_0.4_Bi_1.6_O_6_ (M = La, Ce, Pr, Pb, Y) photoelectrodes were characterized by scanning electron microscopy (SEM) (Apreo, Menlo Park, CA, USA). The photocurrent performance was tested with an electrochemical potentiostat (Metrohm, Autolab/PGSTAT 128N, Herisau, Switzerland) connected to a gas-tight homemade 3-electrode measuring cell with a quartz window, under the AM 1.5G illumination modulated by a solar simulator (Enlitech, Kaohsiung, Taiwan, SS-F5-3A) equipped with a 300W Xe lamp (Enlitech, VE1558). The illumination intensity of the solar simulator was calibrated to 100 mW cm^−2^ using a standard solar cell (Enli/SRC2020, SRC-00149) and a Keithley 2400 source meter. In all involved illuminated measurements, photoelectrodes were in a front-illumination configuration. An Ag/AgCl (1M KCl) electrode and a Pt spiral wire were employed as the reference electrode and the counter electrode, respectively. A buffer solution containing 0.1 M Na_2_HPO_4_ and 0.1 M NaH_2_PO_4_ was used as the testing electrolyte (pH ≈ 6.75) (Consort P901). The scan rate for linear sweep voltammetry (LSV) was set at 10 mV s^−1^, with intermittent periodic irradiation. The same scan rate of 10 mV s^−1^ was used for cyclic voltammetry (CV) measurements. Electrochemical impedance spectra (EIS) of the photoelectrodes were recorded with the FRA 32M module (Metrohm, Autolab/PGSTAT 128N), in the dark, in the frequency range of 1Hz to 100 kHz, with an amplitude of 10 mV, in the open-circuit potential (OCP) condition. A 4-electrode cell configuration that introduced a second Pt wire was adopted; it was coupled to the above reference electrode and a 1 μF shunting capacitor to stabilize the circuit and to filter random defect signals from the potentiostat at high frequencies (>10 kHz) caused by the phase delay [31]. Mott–Schottky (M-S) spectra were obtained with the 3-electrode measuring cell in the reverse biased potential window of −0.225 V_RHE_ to 0.3 V_RHE_, in the dark, with an amplitude of 10 mV and a frequency of 1 kHz. The scan started from potentials more cathodic of the OCP and was toward the OCP to several hundred millivolts before approaching OCP [21]. The potential scale versus Ag/AgCl (V_Ag/AgCl_) was converted to the reversible hydrogen electrode scale (V_RHE_) using the Nernst equation, V_RHE_ = V_Ag/AgCl_ + 0.059 × pH + 0.23.

## 3. Results and Discussion

### 3.1. Crystal Structure

The crystal-structure information of Ba_2_M_0.4_Bi_1.6_O_6_ (M = La, Ce, Pr, Pb, Y) powders was obtained by XRD measurements. In Figure 1a, diffraction peaks appearing at around 2*θ* ≈ 20.2°, 28.8°, 41.2°, 51.1°, 59.7°, and 67.6° are characteristic of the double-perovskite phase [20], confirming the formation of double perovskites in the as-prepared powder samples. A clear trend of diffraction peaks shifting slightly toward higher angles is observed in Figure 1 for Ba_2_M_0.4_Bi_1.6_O_6_ double perovskites with doping ions from La^3+^ to Pb^4+^, which is consistent with the results from Matsushita et al. [19] and could be related to the decrease in ionic radii of the dopant ions and the lattice parameters [32,33], as shown in Table 1. As it is well-established in the literature that Pr dopants exist in the mixed valence state of Pr^3+^ and Pr^4+^ for the Ba_2_BiPrO_6_-based compounds [34,35,36], the detailed geometrical tolerance factors (*t*) and octahedral factors (*μ*) based on the substitution ions of M = La, Ce, Pr, Pb, and Y in the double-perovskite compositional formula Ba_2_M_0.4_Bi_1.6_O_6_ are listed in Table 1. As the introduced dopant ions have different ionic radii compared to the original Bi^3+^ ions, 0.76 Å, and Bi^5+^ ions, 1.03 Å, in the octahedra of the parent BBO compound [37], the double-perovskite structure would be directly affected by doping. The calculated values of *t* fall within a range from 0.927 to 0.938, while the values of *μ* vary from 0.640 to 0.621. This indicates structural stability and possibly a decreasing trend in structural distortion from M = La to M = Y [34,38,39].

For Ba_2_M_0.4_Bi_1.6_O_6_, the monoclinic space group *I*2*/m* has been consistently reported when M = La, Ce, Pr, and Pb [34,40,41,42], while the double perovskite has been considered to adopt a cubic space group *Fm*
$\bar{3}$*m* in the case of M = Y [43]. The experimental diffraction data were analyzed with these reported space groups. Structure refinements and lattice parameter calculations with the Rietveld method show reliable results and are listed in Table 2. The Rietveld refinement profile for each sample is shown in Appendix A. The experimental peak splitting at high diffraction angles is pronounced, as shown in the enlarged detailed window of 40° < 2*θ* < 62° in Figure 1b. For M = La, Ce, Pr, and Pb, the diffraction response at around 2*θ* ≈ 41° consists of two closely adjacent peaks, indexing to the monoclinic (220) and (004) planes as the result of peak splitting from the parent cubic (400) plane. In addition, the broad diffraction response at around 2*θ* ≈ 60° is composed of four individual peaks. These sub-peaks can be assigned to the reflection from lattice planes (400), (22$\bar{4}$), (040), and (224), respectively, and are consistent with the diffraction signals of space group *I*2*/m* [40]. For M = Y, unlike the first set of diffraction peaks that are systematically located at relatively lower angles, the second set of diffraction peaks is not splitting even at high 2*θ* angles, thus suggesting a cubic structure. This is a clear sign for a mixture of the cubic phase and the monoclinic phase [44]. The second set of higher-angle peaks at around 2*θ* ≈ 17.9°, 20.7°, 29.4°, 42.2°, 52.2°, 61.1°, and 69.3° corresponds to the reflections from planes (111), (200), (220), (400), (422), (440), and (620) in the cubic phase space group *Fm*
$\bar{3}$*m* [43]. The coexistence of another double-perovskite phase due to partial Bi^5+^ reduction in air during thermal treatment was considered to be the explanation for the additional set of diffraction peaks at lower angles [43]. Considering the high similarity of the first set of peaks with the diffraction of the parental BBO compound and the La-, Ce-, Pr-, and Pb-doped BBO compounds, the monoclinic space group model *I*2*/m* is applied to analyze the phase structure [41]. The refined lattice parameters from the monoclinic BBO *I*2*/m* model are consistent with the earlier reported data [33]. The refined lattice parameters for Ba_2_M_0.4_Bi_1.6_O_6_ when M = La, Ce, Pr, and Pb tend to decrease as the dopants’ ionic radii shrink, which is well in line with the literature and the abovementioned experimental diffraction data [19,33,40].

### 3.2. Optical Properties and Band Structure

The optical light absorption properties of Ba_2_M_0.4_Bi_1.6_O_6_ (M = La, Ce, Pr, Pb, Y) double perovskites are shown in Figure 2a. The samples exhibit strong light absorption in the spectrum from 300 nm to around 1000 nm, with two distinct absorption regions. The absorption region at higher energy from 300 nm to around 500 nm is due to multiple transitions from the valence band (VB) to the inner conduction band (CB), while the absorption region from around 600 nm to higher wavelengths is considered to be related to the transition from the top of VB (VBM) to the bottom of CB (CBM), possibly accompanied by absorption effects from the defects [45]. The absorption edges for M = La and Pr at around 800 nm are in sharp contrast to the long absorption tails extending up to 1000 nm for M = Ce, Pb, and Y. An evaluation of the optical *E*_g_ for samples with M = La and Pr from the Tauc plot in Figure 2b provides estimates of *E*_g_ = 1.57 eV and *E*_g_ = 1.73 eV, respectively, which are close to the reported 1.41 eV~1.5 eV for the parental BBO [21,24], 1.6 eV for Ba_2_LaBiO_6_ [19,20], and 1.1 eV for Ba_2_PrBiO_6_ [19]. The same method has also been conducted for samples with M = Ce, Pb, and Y; see the Tauc-plot overview in Appendix A and the enlarged Tauc plot in Appendix A. It is reported that the extrapolated *E_g_* value from the first linear region corresponds to the optical band gap, which reflects the onset of the optical absorption, while the second linear region corresponds to the energy gap between the inner VB and the inner CB [46,47]. The obtained 0.74 eV < *E*_g_ < 0.8 eV are small and inaccurate due to the lack of a clear absorption edge, which is in line with earlier reports [19,20].

Since the Ba_2_Y_0.4_Bi_1.6_O_6_ sample powders prepared via traditional overnight high-temperature solid-state methods always show an additional double-perovskite phase of BBO, which has been identified through the refinement simulation and calculation above, the light absorption is difficult to assess. Attempts using different sources of starting reagents and further elevating the reaction temperature consistently give rise to phase separation; see Appendix A. The flat absorption tails in Figure 2a are probably caused by a considerable amount of defects relating to structural disorder, such as defect states, mid-gap states, and sub-gap absorption, in the lattice of Ba_2_M_0.4_Bi_1.6_O_6_ when M = Ce or Pb, in line with similar absorption phenomena that were observed in previous investigations [20,48,49,50]. The contribution of defects, e.g., lattice elemental vacancies or anti-sites in Ba_2_M_0.4_Bi_1.6_O_6_ when M = La or Pr to the UV-vis light absorption, could also not be excluded for generating the in-gap states within the band gap of the materials [18,51]. Previous research works have focused on the physical properties of the semiconductor to superconductor transition by monitoring the content of Pb in the Ba_2_Pb*_x_*Bi_2−_*_x_*O_6_ system [32,33,42]. A semiconducting phase has been found for substitution of Bi by Pb with *x* ≤ 1.3, and in the semiconducting phase, a phase transition from the orthorhombic symmetry to the monoclinic *I*2*/m* space group appears for *x* ≤0.4 at room temperature [42]. In addition, a phase transition by decreasing the content of Ce in the Ba_2_Ce*_x_*Bi_2−_*_x_*O_6_ system has been discovered at the boundary *x* value of 0.4, adopting the same monoclinic *I*2*/m* symmetry [41]. The similarity in crystal structure of Ba_2_M_0.4_Bi_1.6_O_6_ when M = Ce and Pb could both cause crystal defects by introducing partial disorder compared to the ordered B1-O-B2 arrangement of the BBO compound [10,44], as discussed below. Sleight and Mattheiss have explained the electronic mechanisms of the superconductivity to semiconductivity transition in the Ba_2_Pb*_x_*Bi_2−_*_x_*O_6_ system [32,33]. It has been noted that strong interactions between overlapping (Pb and Bi) 6s and O 2p states result in a filled single broad CB for the superconductivity. The strong coupling of the CB and the extended O due to the breathing mode of the O octahedra narrows the 6s band, leading to the splitting of the Bi 6s band into two sub-bands and opening up a small band gap adjacent to the Fermi level, *E*_F_, for the end member parental compound BBO [11,41]. In studies on the electronic structures of the semiconducting BBO-based double perovskites, the CB of the parental BBO is mainly the result of hybridization between the Bi^5+^ 6s orbitals and the O 2p orbitals, where the empty Bi^5+^ 6s orbitals form the CBM [18,22]. In the VB, the core levels from the Ba, Bi, and O constituent elements are at low energy, locating at very deep positions [13]. It is found that the filled antibonding states formed by the strong interaction between the Bi^3+^ 6s^2^ lone-pair electrons and O 2p states near the *E*_F_ form the VBM [24].

Doping external cations in the parental BBO compound brings about changes in the composition or shape of the CB and the VB [20,33,45]. In the case of doping with a large amount of Pb in a Ba_2_Pb*_x_*Bi_2−_*_x_*O_6_ system, when *x* > 1.3, there is no band gap, as the phase is superconducting with contracted unit cells, and the breathing mode of the semiconducting B1-O-B2 distortion is suppressed [32,33]. When the Pb-doping amount is decreased to *x* = 0.4, it is reasonable to infer that the *E*_g_ should be between *E*_g_ ≈ 0 for the superconducting phase and 1.48 eV for the parental BBO, due to the declining band-splitting effects from semiconducting BBO to the evolution of the superconducting composition [11,26,33,41]. Illustrations of the electronic-band energy levels are represented in Figure 3. Doping with M = Ce for Ba_2_M_0.4_Bi_1.6_O_6_, which depopulates both the B1 and B2 sites in the structure, analogous to doping with M = Pb, will narrow the Bi^3+^ 6s^2^ band at the VBM and the Bi^5+^ 6s band at the CBM, leading to an increased distance between the VBM and CBM for an enlarged *E*_g_ [41]. Since Ce and Pr dopants are neighboring elements in the periodic system, they have similar electronic structures, Ce ([Xe]4f^1^5d^1^6s^2^) and Pr ([Xe]4f^3^6s^2^). The 5d and 6s orbitals in the Ce-based and the Pr-based oxides are found to be higher in energy than those of the 4f orbitals, which lie above the O 2p orbitals in a VB [52]. In addition, the unoccupied 4f orbitals of the tetravalent Ce^4+^ ([Xe]4f^0^5d^0^6s^0^) are reported to contribute significantly to the CB [53] and are located near the *E*_F_ in the BBO host material [20]. The Pr ions in the Ba_2_M_0.4_Bi_1.6_O_6_ compound are both in the trivalent Pr^3+^ ([Xe]4f^2^6s^0^) and tetravalent Pr^4+^ ([Xe]4f^1^6s^0^) states, and the 4f band is supposed to be split into two sub-bands by the Coulomb correlation [34]. The empty 4f sub-band contributes to the CB and the partially filled 4f sub-band contributes to the VB, with the filled 4f band located above the O 2p band and a possible partial hybridization with the O 2p states [54,55]. It is also mentioned that the 4f states generally shift down to lower levels as the atomic number increases in the rare-earth group; thus, the introduced empty 4f states by Pr could be closer to the CBM than those introduced by Ce doping [20,54]. Band structure calculations revealed that upon incorporating Pr into BBO with a Pr:Bi ratio of 1:1, the original VBM derived from Bi^3+^ 6s^2^ of BBO vanished and the new VBM shifted down toward lower energy. This confirms that the Bi^3+^ ions provide higher energy levels near the top of the VB than the dopant M = Pr, and, thus, the VBM of the partially doped compound Ba_2_Pr_0.4_Bi_1.6_O_6_ is determined by the existing Bi^3+^ [56]. While fully substituting the Bi^5+^ with Sb in the next step, the original CBM derived from the Bi^5+^ 6s orbitals of BBO disappeared, and the new CBM shifted up to a higher level, which also demonstrates that the CBM is determined by the Bi^5+^ ions rather than the dopant Pr [56]. As a result of reducing the amount of Bi^3+^ ions or Bi^5+^ ions in BBO, doping M = Ce and Pr would both lead to an increased *E*_g_. In contrast, for the first member of the rare-earth group, La ([Xe]5d^1^6s^2^), the 4f shell is not occupied due to its high energy [20], while the outer 5d and 6s orbitals are filled by three valence electrons for a low-energy and stable-electronic configuration. It has been found that the unoccupied 6s orbitals from the trivalent La^3+^ ions are at higher levels than the empty 5d orbitals [57], while the transition metal valence d orbitals are also higher in energy compared to the Bi^5+^ 6s orbitals [21,22,58]. When partially doping trivalent ions La^3+^ in the B1-site for the substitution of Bi^3+^ ions, the density of states near the edge of the VB could be diluted, and the strong Bi^3+^ 6s^2^-O 2p antibonding effects on the VBM of BBO host material are reduced, leading to a lower position of the VBM [58,59]. The CBM character is predominantly due to the empty Bi^5+^ 6s orbitals and remains intact, thereby leading to the formation of a large *E*_g_ of the material [58]. In the case of M = Y doping, despite the changes in the fine structure of the VB, the same reasoning as for M = La doping accounts for the lowering of the VBM. Electronic transitions are found possible both to the empty Bi^5+^ 6s states and to the empty Y^3+^ 4d orbitals in the doped sample, as these two orbitals are thought to be close in energy [45]. As a result, an enlarged *E*_g_ relative to the BBO is observed. In short, compared to the parental BBO compound, doping M = Pb for Ba_2_M_0.4_Bi_1.6_O_6_, in principle, would lead to a smaller *E*_g_, while doping M = La, Ce, Pr, or Y would cause a larger *E*_g_, regardless of the extrinsic defects (see Figure 3).

### 3.3. Photoelectrodes Properties

Following structure determination via powder XRD and refinement through simulations, the XRD characterizations of Ba_2_M_0.4_Bi_1.6_O_6_ (M = La, Ce, Pr, Y, Pb) film photoelectrodes prepared with the dip-coating method are additionally conducted [60], and the results are shown in Figure 4. The diffraction pattern of the bare Au substrate is included as a reference. The diffraction peaks occurring in all photoelectrodes at 2*θ* ≈ 38.3° and 2*θ* ≈ 44.5° are ascribed to the signals from the substrate, while the large peaks located at 2*θ* ≈ 20.2°, 28.8°, and 41.2° are indexed to the characteristic peaks from the double perovskites. The enlarged view in Figure 4b exhibits two distinct sets of diffraction features appearing at around 17°~18°. In an ideal cubic double-perovskite *Fm*
$\bar{3}$*m* symmetry, the reflection of the (111) plane is supposed to give rise to a diffraction peak in this region [19,43]. Meanwhile, in the case of doping cations with different ionic radii, the ideal arrangement of octahedra is slightly distorted, resulting in lower symmetry space groups [19,34]. For Ba_2_M_0.4_Bi_1.6_O_6_, the observation of the (111) reflection when M = Y agrees well with the cubic *Fm*
$\bar{3}$*m* symmetry. The lower symmetry *I*2*/m* space group has been resolved for M = La, Ce, Pr, and Pb [34,40,41,42], but only samples with M = La and Pr show pseudocubic (111) reflection peaks, as can be seen in Figure 4b. The same observation has been discussed in earlier reports; the disappearance of this superlattice signal is considered to be the result of B-sites cations disorder [19,34,40]. For M = Ce and Pb, the disorder can involve both B1- and B2-sites, as the B-sites are all equivalent in BaCeO_3_ and BaPbO_3_ perovskites [41,44]. The tetravalent Ce^4+^ and Pb^4+^ ions, thus, could equally occupy both the B1-site of Bi^3+^ and the B2-site of Bi^5+^ in the parental BBO, suppressing the commensurate CDW and causing disorder in the B1-O-B2 framework [19,41]. When M = Pr, it is believed that half of the dopant Pr^3+^ ions are oxidized into Pr^4+^ ions, and 75% of the B-sites cation couples Pr^3+^/Pr^4+^ and Bi^3+^/Bi^5+^ are ordered [34,40], which can explain the experimental observation of the (111) reflection. The alternating arrangements of Ba_2_M_0.4_Bi_1.6_O_6_ double-perovskite octahedra with M = La, Ce, Pr, Pb, and Y doping in the BBO parental material are illustrated and visualized in Figure 5.

The surface morphologies of Ba_2_M_0.4_Bi_1.6_O_6_ (M = La, Ce, Pr, Pb, and Y) photoelectrodes prepared by the dip-coating method were characterized by SEM, as shown in Figure 6. The photoelectrode with dopant M = La consists of micro-grains with an average diameter of approximately 300 nm accumulated together. This La^3+^-doped photoelectrode, serving as the representative of all variants fabricated via four standard dip-coating–calcination cycles, exhibits a characteristic thickness of around 7.7 μm, as estimated from the cross-sectional image (Figure 6b). Similar compositional microparticle structures can also be observed when doping M = Ce or Y in Figure 6c,f, with average sizes of around 130 nm for M = Ce and 100 nm for M = Y. The trend of grain size reduction from doping M = La to M = Ce and Y is consistent with the full width at half maximum (FWHM) observed in the diffraction peaks in Figure 4. In contrast, when doping M = Pr or Pb, tightly packed crystal domains appear instead of micro-grain particles. The size of domains for photoelectrode doping with M = Pr varies from roughly 260 nm to 2.4 μm. The domain sizes for photoelectrodes with M = Pb are generally larger compared to those of photoelectrodes with M = Pr, ranging from at least 1 μm to 6.4 μm. Interestingly, the fine structure of the domains from photoelectrodes with M = Pb can be clearly identified, and are composed of closely stacked two-dimensional layered crystal planes. This is supported by the preferred plane orientation, with diffraction peaks at around 2*θ* ≈ 20.3°, indexed to the (110) or (002) plane, and 2*θ* ≈ 41.3°, indexed to the (220) or (004) plane, as shown in Figure 4a.

The LSV measurements for the Ba_2_M_0.4_Bi_1.6_O_6_ photoelectrodes with M = La, Ce, Pr, Pb, and Y are conducted under chopped 1 sun illumination in a neutral 0.1 M sodium phosphate buffer electrolyte. As shown in Figure 7, the photoelectrodes Ba_2_M_0.4_Bi_1.6_O_6_ with M = La exhibit much better LSV performance compared to the photoelectrodes with M = Ce, Pr, Pb, and Y. The extracted photocurrent density of the Ba_2_M_0.4_Bi_1.6_O_6_ with M = La prepared by the spin-coating method reaches around 1 mA cm^−2^ at 0.3 V_RHE_. The LSV performance is further significantly enhanced for the photoelectrode with M = La prepared by the dip-coating method, which is reflected by an extracted photocurrent of 2.3 mA cm^−2^ at 0.2 V_RHE_. These exceptionally large photocurrents not only demonstrate outstanding PEC performance, but also substantially exceed the previously reported values for similar perovskite oxide photoelectrode systems, such as 0.016 mA cm^−2^ at −0.64 V_RHE_ in Rh:SrTiO_3_ photoelectrode [61], 0.161 mA cm^−2^ at 0.43 V_RHE_ in LaFeO_3_ photoelectrode [62], 0.1 mA cm^−2^ at 0 V_RHE_ in Ba_2_Bi_1.4_Nb_0.6_O_6_ photoelectrode [21], and 0.065 mA cm^−2^ at 0 V_RHE_ in Ba_2_Bi_1.6_Ta_0.4_O_6_ photoelectrode [26]. According to the AM 1.5G illumination (100 mW cm^−2^) that is calibrated with a standard solar cell, the theoretical maximum photocurrent density achievable is 46.3 mA cm^−2^ through numerical integration of the photon flux from the wavelengths of 280 nm to 1200 nm. For the spectral range from 280 nm to 800 nm, which is reflected in the light-absorption spectrum of the La^3+^-doped double perovskite in Figure 2, the theoretical maximum photocurrent density is calculated to be 27.23 mA cm^−2^. The spin-coated Ba_2_La_0.4_Bi_1.6_O_6_ photoelectrode achieves a photocurrent density of 1 mA cm^−2^ at 0.3 V_RHE_, corresponding to a maximum IPCE of around 3.67%, while the dip-coated Ba_2_La_0.4_Bi_1.6_O_6_ photoelectrode demonstrates significantly enhanced performance, with 2.3 mA cm^−2^ at 0.2 V_RHE_, yielding an IPCE of around 8.45%.

In other photoelectrodes with M = Ce, Pr, Pb, and Y, the observed photocurrents are negligible. Considering that the number of photons which can be absorbed to excite the e^−^-h^+^ pairs is determined by the material *E*_g_, the previous light-absorption spectra and the electronic band structure analysis results would suggest comparable LSV performances. This is in line with the LSV photocurrent curves, whose magnitudes are roughly in the range of 2 mA cm^−2^ to 3 mA cm^−2^ at 0.2 V_RHE_ when doping with M = La, Ce, Pr, and Pb, as can be seen in Figure 7. However, an inferior PEC performance is anticipated for the M = Pb photoelectrode with the narrowed *E*_g_ (˂1.5 eV), due to factors such as thermalization losses, the reduced photovoltage, and the rapid charge recombination [63,64]. The large spikes that are detected in the photocurrent curve for doping with M = Y can be attributed to severe photon-excited e^−^ trapping and charge recombination losses due to defects associated with the phase separation [65]. The dark currents for photoelectrodes doped with M = Ce, Pr, Pb, and Y are dominating the overall trends of the LSV patterns, indicating that the dark electrochemical reduction reactions are playing a significant role. Dark electrochemical corrosion currents are also observed in the dark LSV curves in Figure 7a for the photoelectrode doped with M = La, with a broad shoulder peak from potentials around 0.5 V_RHE_ to 0.75 V_RHE_ and an abrupt large reduction peak roughly from 0.2 V_RHE_ to lower potentials. The first broad reduction peak in the range from 0.5 V_RHE_ to 0.75 V_RHE_ could be ascribed to the reduction reactions of Bi^5+^ → Bi^3+^ ($E_{{Bi}^{5+}/{Bi}^{3+}}^{0}$ = 1.8 V_SHE_) and the processes of Bi^5+^ → Bi^4+^ → Bi^3+^ ($E_{{Bi}^{5+}/{Bi}^{4+}}^{0}$ = 2 V_SHE_,
$E_{{Bi}^{4+}/{Bi}^{3+}}^{0}$ = 1.59 V_SHE_) [66]. The strong reduction peak at 0.2 V_RHE_ to increasingly cathodic potentials can be attributed to reduction reactions of ionic Bi^3+^ species [66,67].

These reduction observations align with the CV patterns of the parent BBO thin-film photoelectrodes, as shown in Appendix A, for both cathodic and anodic scan directions. Notably, the undoped BBO photoelectrodes are prone to predominant decomposition within the potential range of 0.5 V_RHE_ to 0.8 V_RHE_, with the main reduction processes being irreversible. Since the reduction potentials for La^3+^ and Y^3+^ are rather low (below −1.7 V_SHE_), they are supposed not to contribute to the Faradaic signals for electrochemical corrosion in the experimental potential range in Figure 7. In contrast, the redox potentials of Ce^4+^ ($E_{{Ce}^{4+}/{Ce}^{3+}}^{0}$ = 1.72 V_SHE_), Pr^4+^ ($E_{{Pr}^{4+}/{Pr}^{3+}}^{0}$ = 3.2 V_SHE_), and Pb^4+^ ($E_{{Pb}^{4+}/{Pb}^{2+}}^{0}$ = 1.7 V_SHE_) are relatively high, and conversions are thermodynamically facile in the operational potential window [66]. Hence, additional reductions of these dopant ions in the photoelectrodes can cause critical decomposition of the double-perovskite materials. This is consistent with the observed corrosion reaction peaks that show up at potentials from 0.2 V_RHE_ to 0.4 V_RHE_ for the photoelectrode with M = Ce in Figure 7b and in the potential range from 0.8 V_RHE_ to 1 V_RHE_ for the photoelectrode with M = Pb in Figure 7c. In Figure 7a, the dark electrochemical reduction peak from 0.5 V_RHE_ to 0.75 V_RHE_ disappeared for the 7.7 μm film photoelectrode with M = La prepared by the dip-coating method, signifying an improved electrochemical corrosion resistance. In the potential region from 0.2 V_RHE_ to lower values, serious electrochemical corrosion occurs, despite the photoelectrode showing a decent net photocurrent response of 1.5 mA cm^−2^ at 0 V_RHE_. The thickness of the thin film photoelectrode with M = La fabricated via the 4-times spin-coating–calcination procedure was examined by the SEM cross-sectional image, which reveals a thickness of around 1 μm; see Appendix A. This makes it clear that increasing the thickness of the photoelectrode with M = La is beneficial for improving the electrochemical stability, while depositing a thin protection overlay could be helpful to alleviate the deterioration of the photoelectrode for superior LSV performance and proton-reduction applications [27].

The surface morphologies of the dip-coated Ba_2_M_0.4_Bi_1.6_O_6_ (M = La, Ce, Pr, Pb, and Y) photoelectrodes after LSV measurements with periodic illumination are presented in Appendix A. It is clear that a large number of flower-like 3D microstructures with sizes below 5 μm are formed on the surface of the photoelectrode with M = La. Compared to the initial neat surface of the photoelectrode with M = Pr, additional irregular particles and flat flower-like microstructures are found on the surface after the LSV measurements. For the photoelectrode with M = Pb, the original ordered layered structure on the surface is replaced by abundant hollow spheres and sea urchin-like complex structures. In contrast, on the surfaces of the photoelectrodes with relatively low crystallinity when doping M = Ce and Y, only irregular sub-micron particles are formed. The significant visible changes on the Ba_2_M_0.4_Bi_1.6_O_6_ (M = La, Ce, Pr, Pb, and Y) photoelectrodes’ surfaces indicate the decomposition of the electrode after LSV tests.

The EIS patterns of the well-performing photoelectrodes Ba_2_M_0.4_Bi_1.6_O_6_ with M = La are shown in Figure 8a. For comparison, the EIS tests were also conducted for photoelectrodes BBO and Ba_2_M_0.4_Bi_1.6_O_6_ with M = Pb prepared by 4-times spin-coating. The photoelectrode Ba_2_M_0.4_Bi_1.6_O_6_ with M = La fabricated by 4-times dip-coating (~7.7 μm) is apparently thicker than the photoelectrode fabricated by 4-times spin-coating, which is around 1 μm, as indicated in the cross-section image of the photoelectrode in Appendix A. The equivalent circuit, which fits and simulates the EIS data, consists of two Randles circuits (RCs) connected to a series resistance, *R*_s_ [23,68]. The *R*_ct_ and *C*_dl_ in the first RC represent the charge transfer resistance and the capacitance of a constant phase element extended from the double-layer capacitance on the electrolyte side at the photoelectrode/electrolyte interface, respectively. The radii of the correlated large semicircles in the medium-to-low frequency range in the EIS patterns in Figure 8 signify the magnitude of the *R*_ct_. The *R*_sc_ and *C*_sc_ in the second RC are the coupled resistance and capacitance from the semiconductor photoelectrodes, which correspond to the small semicircles in the high frequency region of the spectra. Parameters of the equivalent circuit after fitting and modeling are listed in Table 3.

The radius of the arc at high frequency for the thicker photoelectrode, Ba_2_M_0.4_Bi_1.6_O_6_, with M = La, prepared by the dip-coating method, exhibits a large *R*_sc_ of around 1.2 kΩ, while the thinner photoelectrode with the same composition prepared by the spin-coating method shows a smaller *R*_sc_ of around 1.1 kΩ, indicating an increase in the ohmic resistance of the semiconducting photoelectrode for M = La when increasing the film thickness. In sharp contrast, the *R*_sc_ of the thin-film photoelectrodes for M = Pb and the parental BBO prepared by the same spin-coating method exhibit significantly reduced values, by at least an order of magnitude, roughly in a range from tens to one hundred ohms. This observation is in line with the analysis results above that doping M = La causes an increased *E*_g_ in comparison with the semiconducting parental BBO compound and that doping M = Pb in BBO is helpful for reducing the *E*_g_ as a favorable strategy for producing BBO-based superconductors [32]. The photoelectrode with M = Pb shows an *R*_sc_ at a similar level to BBO, yet slightly higher, which can be attributed to bulk trapping states arising from crystal lattice disorder defects that hinder charge transport within the electrode [69]. For Ba_2_M_0.4_Bi_1.6_O_6_ photoelectrodes with M = La, the thinner spin-coated sample shows a larger *C*_sc_ than the thicker dip-coated one, which could be the result of additional contributions from larger quantities of surface states due to the surface crystal defects, including the semiconductor depletion layer [68]. The parental compound BBO shows the highest *C*_sc_ compared to the doped ones, which can be strongly correlated to crystal defects, i.e., oxygen vacancies and Bi vacancies, caused by high-temperature treatments [18,21]. When assessing the charge transfer properties at the photoelectrode/electrolyte interface, the *R*_ct_ for the parental BBO displays a higher value than the M = La-doped ones. This is consistent with the improved catalytic activity of the BBO-based double-perovskite materials by doping rare-earth elements for better interface charge transport [20]. Similarly, with M = Pb, the small *R*_ct_ for the photoelectrode can be directly attributed to the introduction of Pb^4+^ ions, which likely exhibit kinetically rapid catalytic activities [70]. In addition, the contraction of unit cells with smaller lattice parameters when doping M = Pb appears to be closely related to fast electron hopping and a thermodynamically favorable redox potential of Pb^4+^ ions [41,66].

Mott–Schottky (M-S) characterization was carried out to further gain insights into the Ba_2_La_0.4_Bi_1.6_O_6_ photoelectrodes (Appendix A). The negative slopes of the M-S curves confirm the p-type conductivity of the semiconductor photoelectrodes [21]. The M-S approximation,
(1)$$\frac{1}{C_{sc}^{2}}=\frac{2}{\varepsilon_{r}\varepsilon_{0}eN_{a}}\left(E_{app}-E_{fb}-\frac{k_{B}T}{e}\right),$$ is mostly used to simplify the relationship between the semiconductor capacitance, *C*_sc_, and the applied potential, *E*_app_, where
$\varepsilon_{r}$ is the relative dielectric constant of the semiconductor,
$\varepsilon_{0}$ is the vacuum permittivity, *e* is the electronic charge, *N*_a_ is the acceptor concentration, *E*_fb_ is the flat band potential, *k*_B_ is the Boltzmann constant, and *T* is the absolute temperature [21]. This characteristic expression is validated based on the pre-assumption that the potential drop across the photoelectrode/electrolyte junction, as a function of applied bias potential, is primarily governed by the semiconductor potential drop,
$\left|∆\emptyset_{sc}\right|$, rather than the electrolyte double-layer potential drop,
$\left|∆\emptyset_{dl}\right|$, namely
$\left|∆\emptyset_{sc}\right|\gg \left|∆\emptyset_{dl}\right|$ and
$C_{sc}\ll C_{dl}$; thus,
$\frac{1}{C_{interface}}\approx \frac{1}{C_{sc}}$ [71,72]. From the EIS analysis above, the *C*_SC_ and the *C*_dl_ are in the same order of magnitude both for the dip-coated and the spin-coated Ba_2_La_0.4_Bi_1.6_O_6_ photoelectrodes; thus, the measured total interfacial capacitance contribution should include the term relating to the *C*_dl_ [71,72]:
(2)$$\frac{1}{C_{interface}}=\frac{1}{C_{sc}}+\frac{1}{C_{dl}}$$

The M-S equation with the electrolyte potential drop correction term is then expressed as follows [71,73]:
(3)$$\frac{1}{C_{sc}^{2}}=\frac{2}{\varepsilon_{r}\varepsilon_{0}eN_{a}}\left(E_{app}-E_{fb}-\frac{k_{B}T}{e}\right)+\frac{1}{C_{dl}^{2}}$$

The interfacial potential drop apparently originates from both the space charge region of the solid semiconductor and the electrical double layer in the liquid electrolyte. Considering the material
$\varepsilon_{r}$ of the two boundary members, the BBO and the Ba_2_LaBiO_6_, to be around 31 and 18.7 [74,75], the slopes of the linear region in the M-S plot point to an *N*_a_ of around 1.7~2.8 × 10^19^ cm^−3^ for the dip-coated and 2.7~4.4 × 10^19^ cm^−3^ for the spin-coated Ba_2_La_0.4_Bi_1.6_O_6_ photoelectrode. This order of magnitude in ionized acceptor concentration is high compared to the other homolog doping systems [21,22] and could possibly be part of the reason for the superior high-photocurrent-density performance. The *E*_fb_ can be derived from the intersection on the potential axis based on the following:
(4)$$E_{app}=E_{fb}+\frac{k_{B}T}{e}-\frac{\varepsilon_{r}\varepsilon_{0}eN_{a}}{2}\times \frac{1}{C_{dl}^{2}}$$

And the width of the semiconductor depletion region can be calculated based on the following [73,76]:
(5)$$W=\sqrt{\frac{2\varepsilon_{r}\varepsilon_{0}(E_{app}-E_{fb})}{eN_{a}}}.$$

The obtained *E*_fb_ for the ~1 μm 4T-spin-coated photoelectrode is 0.26 V_RHE_, which is very close to the *E*_fb_ previously reported for a p-type thin-film homolog system [21]. The *E*_fb_ of 0.78 V_RHE_ for the ~7.7 μm 4T-dip-coated photoelectrode also aligns well with the oxygen vacancy defect-regulated n-type counterpart homolog material with the same compound formula [22]. Furthermore, the calculated reasonable space charge-region thickness, *W*, is around 1.6~2.7 nm at 0.2 V_RHE_ for the ~1 μm 4T-spin-coated photoelectrode and around 4.9~8.2 nm at 0.45 V_RHE_ for the ~7.7 μm 4T-dip-coated photoelectrode, which is in line with partial depletion of the semiconductor photoelectrodes during the M-S tests [76]. The *E*_fb_ potentials of 0.26 V_RHE_ and 0.78 V_RHE_ are remarkably close to the apparent dark reduction potentials of 0.2 V_RHE_ and 0.75 V_RHE_ from the LSV tests in Figure 7. Generally, surface states charging and Faradaic redox reactions on the semiconductor photoelectrodes surface cause band-edge shifting and, thus, *E*_fb_ shifting [77,78], leading to the slight deformation of the ideal M-S curve based on the “pinning” band-edge model [72]. The strategic doping approach in this work provides significant insights into the crystal and electronic band structure control of the BBO-based double perovskites, and it opens new avenues for further forwarding this family of novel photoelectrode materials into promising chemical-engineering and sustainable energy-conversion devices.

## 4. Conclusions

In this work, a series of doped Ba_2_M_0.4_Bi_1.6_O_6_ (M = La, Ce, Pr, Pb, Y) double perovskites at the B-sites were prepared via high-temperature solid-solution methods. XRD patterns and the Rietveld refinement calculations confirm that the Ba_2_M_0.4_Bi_1.6_O_6_ double perovskites with M = La, Ce, Pr, and Pb share the monoclinic *I*2*/m* symmetry. In contrast, the Ba_2_M_0.4_Bi_1.6_O_6_ with M = Y shows two mixed double-perovskite phases, with one in the cubic *Fm*
$\bar{3}$*m* space group and the other in the monoclinic *I*2*/m* space group. The lattice parameters of the monoclinic Ba_2_M_0.4_Bi_1.6_O_6_ double perovskites indicate a decreasing trend that aligns with reducing the radii of the B-sites ions. When doping trivalent M = La and Y, the highly ordered B1-O-B2 structure is maintained; meanwhile, when doping tetravalent M = Ce and Pb, the substituted B-sites become disordered. For the mixed trivalent and tetravalent case with M = Pr, partial ordering is achieved with the B-sites mixing. Further improving the sample quality of M = Y could involve creating an oxygen-rich environment and shortening the annealing time, with the aim to mitigate the oxygen vacancies and the loss of Bi caused by high-temperature calcination [18,20,21,43], and ultimately enable the formation of a pure phase compound. All the Ba_2_M_0.4_Bi_1.6_O_6_ samples demonstrate excellent light-absorption capabilities over a broad spectrum, spanning around 1000 nm. The estimated optical *E_g_* from the corresponding Tauc plot indicates a value of around 1.57 eV and 1.73 eV for double perovskites with M = La and M = Pr, respectively. When tested as photoelectrodes, the Ba_2_M_0.4_Bi_1.6_O_6_ with M = La delivers a remarkable photocurrent density of 1 mA cm^−2^ at 0.3 V_RHE_ for the 1 μm thin film fabricated by the 4-times spin-coating process, and a further boosted photocurrent density of 2.3 mA cm^−2^ at 0.2 V_RHE_ for the 7.7 μm thicker film fabricated by the 4-times dip-coating process, outperforming other BBO-based homologous systems due to a combination of strong light absorption and significantly enhanced stability.

## Figures and Tables

**Figure 1 nanomaterials-15-01039-f001:**
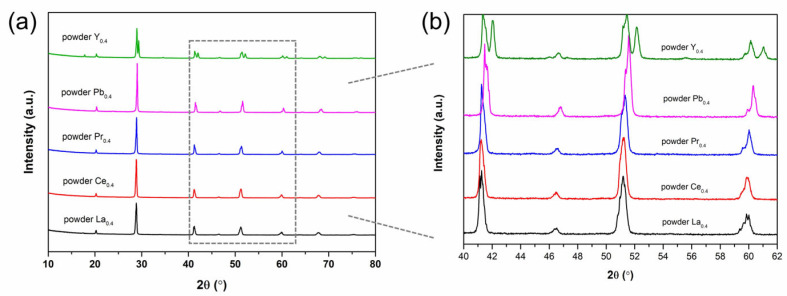
(**a**) XRD patterns of Ba_2_M_0.4_Bi_1.6_O_6_ (M = La, Ce, Pr, Pb, Y) powders, and (**b**) enlarged XRD patterns for the diffraction angle range of 40° < 2*θ* < 62°.

**Figure 2 nanomaterials-15-01039-f002:**
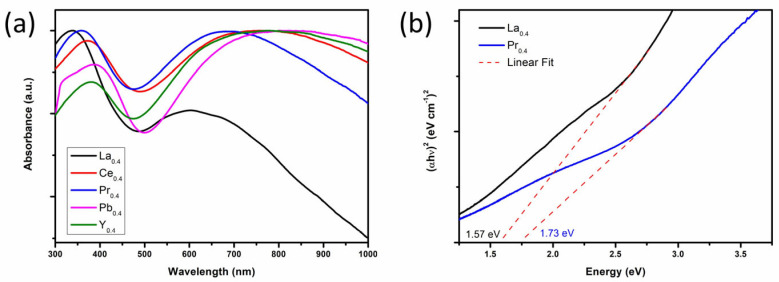
(**a**) UV-vis absorption spectra of powder Ba_2_M_0.4_Bi_1.6_O_6_ (M = La, Ce, Pr, Pb, Y) and (**b**) the corresponding Tauc plot for samples M = La and Pr.

**Figure 3 nanomaterials-15-01039-f003:**
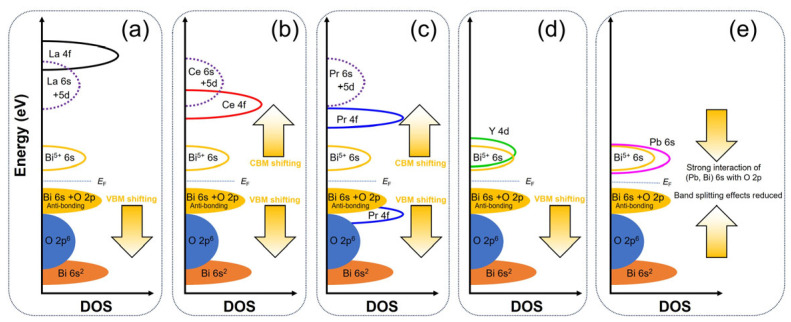
Schematic representation of energy-level diagrams for Ba_2_M_0.4_Bi_1.6_O_6_ with M = (**a**) La, (**b**) Ce, (**c**) Pr, (**d**) Y, and (**e**) Pb.

**Figure 4 nanomaterials-15-01039-f004:**
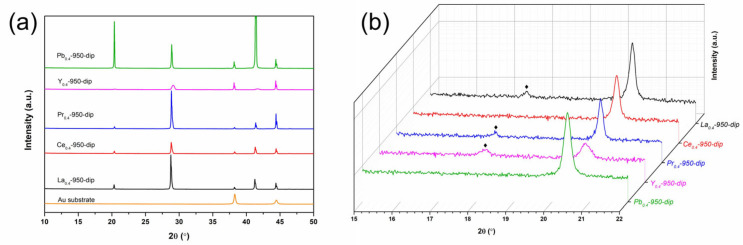
(**a**) XRD patterns of Ba_2_M_0.4_Bi_1.6_O_6_ (M = La, Ce, Pr, Y, Pb) film photoelectrodes prepared by dip-coating and (**b**) the corresponding enlarged XRD patterns in the diffraction angle range 15° < 2*θ* < 22°.

**Figure 5 nanomaterials-15-01039-f005:**
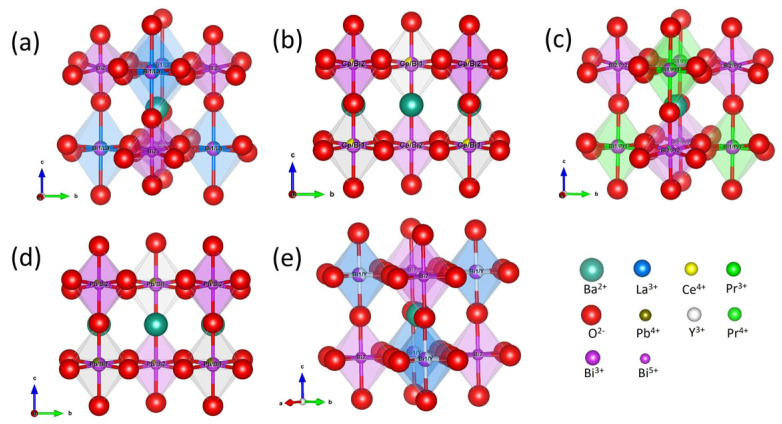
Octahedron arrangement of perovskite structures in Ba_2_M_0.4_Bi_1.6_O_6_ when M = (**a**) La, (**b**) Ce, (**c**) Pr, (**d**) Pb, and (**e**) Y, with Bi^3+^ ions occupying B1-sites and Bi^5+^ ions occupying B2-sites in the host BBO parental perovskite.

**Figure 6 nanomaterials-15-01039-f006:**
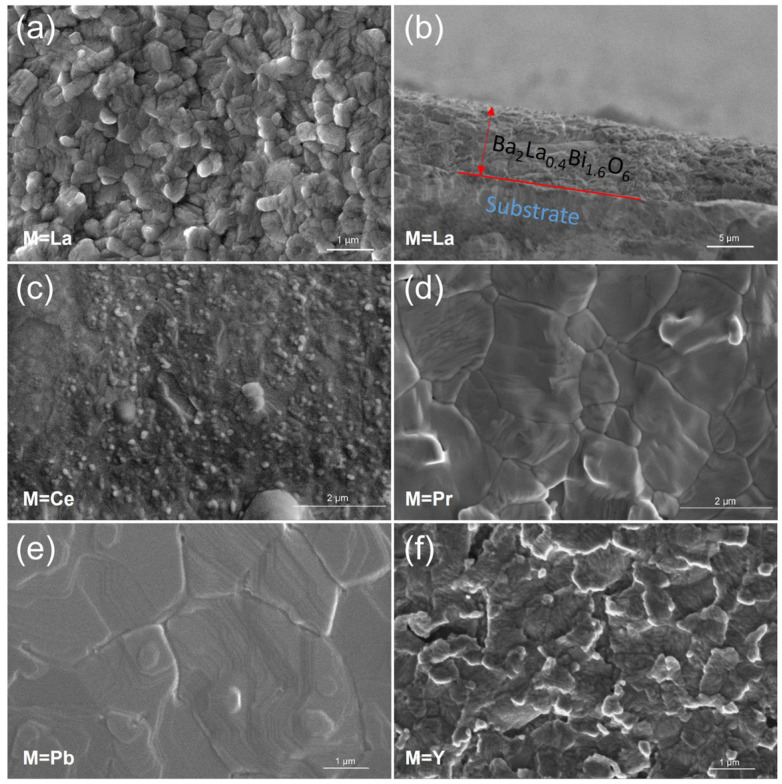
Surface morphologies of the dip-coated Ba_2_M_0.4_Bi_1.6_O_6_ photoelectrodes for M = (**a**) La, (**c**) Ce, (**d**) Pr, (**e**) Pb, and (**f**) Y; and (**b**) the cross-sectional image of the dip-coated photoelectrode Ba_2_M_0.4_Bi_1.6_O_6_ for M = La.

**Figure 7 nanomaterials-15-01039-f007:**
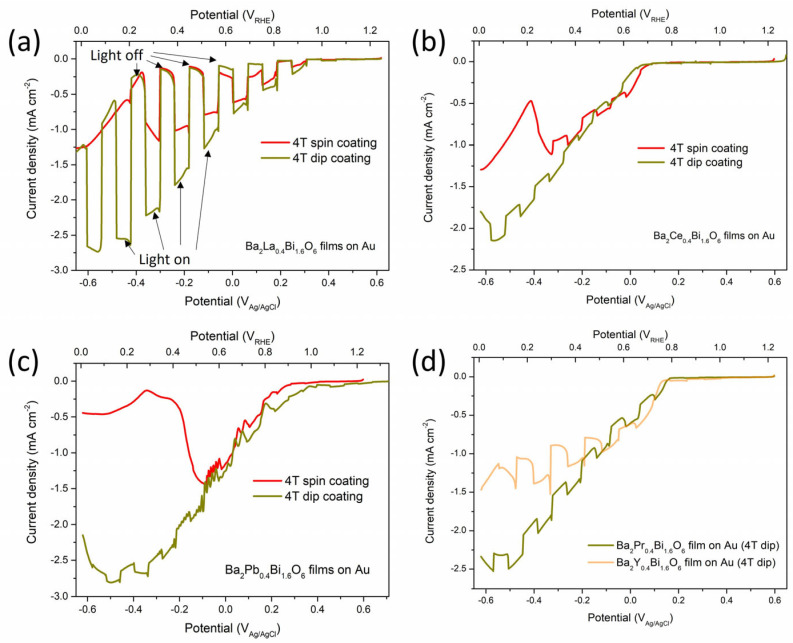
LSV profiles for the Ba_2_M_0.4_Bi_1.6_O_6_ photoelectrodes with (**a**) M = La, (**b**) M = Ce, and (**c**) M = Pb prepared by the spin-coating method or the dip-coating method, and with (**d**) M = Pr and Y prepared by the dip-coating method, in the neutral electrolyte (pH ≈ 6.75) containing 0.1 M Na_2_HPO_4_ and 0.1 M NaH_2_PO_4_ under periodically chopped 100 mW cm^−2^ illumination.

**Figure 8 nanomaterials-15-01039-f008:**
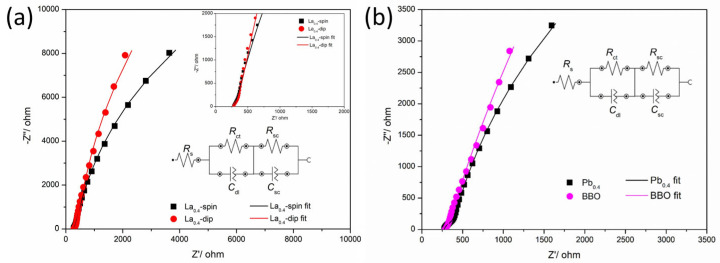
(**a**) The dark EIS curves of the Ba_2_M_0.4_Bi_1.6_O_6_ photoelectrodes for M = La by the spin-coating method (in black) and the dip-coating method (in red); the insert graph shows the magnified spectra at medium-to-high frequencies. (**b**) The dark EIS curves of the spin-coated Ba_2_M_0.4_Bi_1.6_O_6_ photoelectrode for M = Pb and the spin-coated BBO photoelectrode are shown for comparison.

**Table 1 nanomaterials-15-01039-t001:** Calculated tolerance factors *t* and octahedral factors *µ* *.

Ion	La^3+^	Ce^4+^	Pr^3+^	Pr^4+^	Pb^4+^	Y^3+^
Coordination number	6	6	6	6	6	6
Ionic radius (Å)	1.032	0.87	0.99	0.85	0.775	0.9
Formula	Ba_2_La_0.4_Bi_1.6_O_6_	Ba_2_Ce_0.4_Bi_1.6_O_6_	Ba_2_Pr_0.4_Bi_1.6_O_6_		Ba_2_Pb_0.4_Bi_1.6_O_6_	Ba_2_Y_0.4_Bi_1.6_O_6_
*t*	0.927	0.929	0.931		0.937	0.938
*µ*	0.640	0.636	0.633		0.622	0.621

*
$t=\frac{\left(r_{A}+r_{O}\right)}{\sqrt{2}\left(r_{B}+r_{O}\right)}$,
$\mu =\frac{r_{B}}{r_{O}}$, where *r*_A_ is the radius of Ba^2+^ ions at the A-sites, *r*_B_ is the average radius of ions at the B-sites and *r*_O_ is the radius of O^2−^ ions; all radii are the effective Shannon ionic radii [37].

**Table 2 nanomaterials-15-01039-t002:** Lattice parameters of Ba_2_M_0.4_Bi_1.6_O_6_ (M = La, Ce, Pr, Pb, Y) powders obtained via Rietveld refinement.

Sample	La_0.4_	Ce_0.4_	Pr_0.4_	Pb_0.4_	Y_0.4_	
Phase	1	1	1	1	1	2
Space group	*I*2*/m*				*Fm* $\bar{3}$ *m*	*I*2*/m*
*a* (Å)	6.2156 (2)	6.1559 (5)	6.1942 (3)	6.1176 (3)	8.5827 (2)	6.1838 (3)
*b* (Å)	6.1598 (3)	6.1953 (4)	6.1469 (3)	6.1566 (3)		6.1409 (4)
*c* (Å)	8.7137 (4)	8.7045 (6)	8.6898 (4)	8.6401 (4)		8.6775 (6)
*β* (°)	90.305 (4)	90.02 (2)	89.884 (8)	90.0229 (0)	90	90.1599 (0)
*R*_wp_ (%)	5.64	6.93	6.68	7.35	8.69	
χ 2	1.74	2.15	2.01	2.56	2.56	

**Table 3 nanomaterials-15-01039-t003:** Equivalent circuit parameters by fitting and simulating the experimental EIS data.

Sample	*R*_s_/Ω	*R*_ct_/Ω	*C*_dl_/F	*R*_sc_/Ω	*C*_sc_/F	χ 2
La_0.4_-dip	278.3 (±1.1)	47.1 (±3.3) × 10^3^	1.5 (±0.0) × 10^−5^	1203.5 (±18.3)	1.4 (±0.1) × 10^−5^	0.025
La_0.4_-spin	277.0 (±1.7)	28.5 (±1.0) × 10^3^	1.4 (±0.0) × 10^−5^	1100.0 (±10.0)	2.1 (±0.2) × 10^−5^	0.014
Pb_0.4_-spin	274.8 (±0.4)	17.6 (±0.4) × 10^3^	3.1 (±0.0) × 10^−5^	102.0 (±1.4)	1.4 (±0.2) × 10^−5^	0.003
BBO-spin	283.9 (±0.8)	56.5 (±1.4) × 10^3^	2.8 (±0.0) × 10^−5^	27.0 (±2.3)	2.4 (±0.4) × 10^−5^	0.018

## Data Availability

The data are available upon reasonable request from the corresponding author.

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
