# Peer review of "Strategic Doping for Precise Structural Control and Intense Photocurrents Under Visible Light in Ba2M0.4Bi1.6O6 (M = La, Ce, Pr, Pb, Y) Double Perovskites"

_nanomaterials, 2025, doi:10.3390/nano15131039_

Round 1

Reviewer 1 Report

Comments and Suggestions for Authors

The paper presents an interesting approach to synthesizing and using oxide thin-film electrodes based on Ba₂Bi₂O₆ doped with rare-earth elements, but some parts need improvement, especially the consistency between powder and thin-film data, and some missing experimental details. After a major revision, the manuscript could be suitable for re-evaluation.

Q1) If I understood correctly, the main topic of this paper is to employ those compounds in electrodes. There is no reason to show XRD and Absorption spectra for powder samples, while electrodes are made from solution. There is no direct link or transition/explanation in the text. 

Q2) I have some doubts concerning the fittings of absorption spectra. Could the authors show a broader spectrum on the X and Y axes? Please do it in both the main manuscript and the supporting information. 

Q3) There is a lack of Rietveld refinement fitting results for the XRD of the layer sample.

Q4) Powder XRD Rietveld refinement lacks the χ2 values.

Q5) A SEM/TEM cross-section of the electrodes would be beneficial. 

Q6) There is no comparison of the estimated photocurrent density with other literature data. There are only values without discussion if the obtained values are good enough.  

Q7) There is no crucial information about the stability of the electrodes under long light exposure.

Q8) The manuscript lacks crucial information about the IPCE of the obtained electrode. This limits the interpretation of light to charge conversion efficiency. 

Author Response

#reviewer 1

Comments and Suggestions for Authors

The paper presents an interesting approach to synthesizing and using oxide thin-film electrodes based on Ba₂Bi₂O₆ doped with rare-earth elements, but some parts need improvement, especially the consistency between powder and thin-film data, and some missing experimental details. After a major revision, the manuscript could be suitable for re-evaluation.

Q1) If I understood correctly, the main topic of this paper is to employ those compounds in electrodes. There is no reason to show XRD and Absorption spectra for powder samples, while electrodes are made from solution. There is no direct link or transition/explanation in the text.

Response: We apologize for the confusion. The specific purpose of our paper is to investigate the strategic doping, and the electrodes are included mainly for benchmarking of novel materials. The electrodes are made of solid crystalline materials obtained from solution. The characterization of electrodes with solid crystalline materials is performed with XRD to validate the crystallinity, and how this compares with the parent powder materials. This is in line with the current practice in the field of research. In contrast with the XRD, absorption spectroscopy is difficult to perform on the electrodes. We apologize for any confusion caused to the reviewer, and we have now added “To benchmark the solar-to-current conversion performance,……as highly crystalline thin film photoelectrodes” in the materials and methods section on page 3, and “Following structure determination via powder XRD……results are shown in Figure 4.” on page 9, line 352 to 355. To underline that this is an established practice in the field, we have also added a reference to a seminal paper by McCusker et al.( L.B. McCusker, et al., J. Appl. Crystallogr., 1999. 32(1): p. 36-50.), and one to our own work (T. Guo, W.T. Fu and H.J.M. de Groot, Small, 2024. 20: p. 2308781.).

Q2) I have some doubts concerning the fittings of absorption spectra. Could the authors show a broader spectrum on the X and Y axes? Please do it in both the main manuscript and the supporting information.

Response: Thank you for your suggestion. The overview of the full Tauc plots for direct transition is added to the supporting information in Figure S3. This Figure is now referenced in the main text on page see page 7, line 247 to 252, “see the Tauc plot overview in Figure S3,……, which is in line with earlier reports.”

Q3) There is a lack of Rietveld refinement fitting results for the XRD of the layer sample.

Response: The main purpose of XRD is to identify the phase and the purity of films. The XRD data of thin films have low intensity and the diffraction peaks are often broadened due to grain size effects, making it challenging to extract accurate structural parameters through Rietveld refinement. (see: R.A. Young (ed.). (1993). The Rietveld Method. Oxford University Press.)

Q4) Powder XRD Rietveld refinement lacks the χ2 values.

Response: Powder XRD Rietveld refinement χ2 values were already listed at the bottom of Table 2 in the manuscript.

Q5) A SEM/TEM cross-section of the electrodes would be beneficial.

Response: Thank you for your comments. We agree that SEM/TEM cross-sectional images of the electrodes clarify the configurations and assist to understand the physical-chemical properties. For this reason, we show in Figure 6(b) and Figure S7 SEM cross-sectional images of the double perovskite with La3+ doping, as this is the representative and targeting compound with overwhelming photocurrents, outperforming the other counterpart homologs. This is now stated in the manuscript on page 10 line 385 to 388 “This La3+ doped photoelectrode,….., from the cross-sectional image (Figure 6b) ”, and page 13 line 473 to 476 “The thickness of the thin film….,…. a thickness of around 1 μm, see Figure S7”. A full SEM characterization of all the doping systems goes beyond the current study and could be a topic of future study.

Q6) There is no comparison of the estimated photocurrent density with other literature data. There are only values without discussion if the obtained values are good enough.

Response: Thank you for your suggestion. The enhanced photocurrent densities through strategic doping have now been compared to other literature data in the manuscript, on page 12, line 413 to 418 “These exceptionally large photocurrents……,……in Ba2Bi1.6Ta0.4O6 photoelectrode”.

Q7) There is no crucial information about the stability of the electrodes under long light exposure.

Response: As stated in the answer to Q1, the prime purpose of our investigation is the exploration of strategic doping in the Ba2Bi2O6 framework, and the electrodes are prepared for benchmarking the novel materials. A full characterization of electrode systems including long term stability is well beyond this first preparative investigation and can be the topic of future research.

Q8) The manuscript lacks crucial information about the IPCE of the obtained electrode. This limits the interpretation of light to charge conversion efficiency. 

Response: We appreciate your comments. IPCE is a level of detail that we not yet address in the present investigation. Since we have used an AM 1.5G illuminator, we can provide the estimated entire incident photon to current efficiency. According to the AM 1.5G illumination (100 mW cm-2) that is calibrated with a standard solar cell, the theoretical maximum photocurrent density achievable is 46.3 mA cm-2 through numerical integration of the photon flux from the wavelengths of 280 nm to 1200 nm. For the spectral range from 280 nm to 800 nm, which is reflected in the light absorption spectrum of the La3+ doped double perovskite in Figure 2, the theoretical maximum photocurrent density is calculated to be 27.23 mA cm-2. The spin-coated Ba2La0.4Bi1.6O6 photoelectrode achieves a photocurrent density of 1 mA cm-2 at 0.3 VRHE, corresponding to a maximum IPCE of around 3.67%, while the dip-coated Ba2La0.4Bi1.6O6 photoelectrode demonstrates significantly enhanced performance with 2.3 mA cm-2 at 0.2 VRHE, yielding an IPCE of around 8.45%. This is now included in the main text on page 12, line 418 to 427 “According to the AM 1.5G illumination……,……yielding an IPCE of around 8.45%”.

Reviewer 2 Report

Comments and Suggestions for Authors

Author Response

#reviewer 2

This work systematically investigates the structural tuning of Ba2M0.4Bi1.6O6 (M = La, Ce, Pr, Pb, Y) double perovskite materials and their effects on light absorption and photoelectrochemical performance. The experimental design is reasonable, the characterization data are sufficient, and the conclusions have a certain degree of novelty and application potential.

Comment 1:

The manuscript repeatedly mentions that Pr exists in a mixed valence state of Pr³⁺/Pr⁴⁺ and suggests that this mixture contributes to partial B-site ordering. However, no electronic structure data or supporting literature evidence is provided to justify this assumption. It is recommended that the authors clarify the origin of the Pr valence states and provide elemental analysis or valence state measurements to confirm the actual state of the dopant ions.

Response: We appreciate the reviewer’s comments. It is well known for chemical systems that Pr may exist in both Pr3+ and Pr4+electronic configurations. For example, the commercially available oxide has the formula Pr6O11, which is a mixed valence oxide. In perovskites, the oxidation state of Pr depends critically on the presence of other metal ions. Examples are BaPrO3 (Pr4+), Ba2PrIrO6 (Pr4+) (ref: W.T. Fu and D.J.W. Ijdo, J. Solid State Chem. , 2005. 178(4): p. 1312-1316..), PrAlO3 (Pr3+), Ba2PrSbO6 (Pr3+) (ref: W.T. Fu and D.J.W. Ijdo, J. Solid State Chem., 2005. 178(7): p. 2363-2367.). The oxidation states of Pr are well defined by the bond valence sum parameters in these compounds. In addition, it is also well-established in the literature that Pr dopants exist in the mixed valence state of Pr3+and Pr4+ for the Ba2BiPrO6-based compounds (ref: W.T.A. Harrison, et al., Chem. Mater., 1995. 7(11): p. 2161-2167, and V. Poltavets, P. Kazin and M. Jansen, Solid State Sci., 2006. 8(10): p. 1152-1159 on mixed valence state ; ref : A. Sato, et al., Solid State Sci., 2020. 107: p. 106352 on electronic structure).

This is now included in the main text on page 5 line 193-194, “As it is well-established in the literature that Pr dopants exist in the mixed valence state of Pr3+ and Pr4+ for the Ba2BiPrO6-based compounds”. We have now also added all relevant supporting references to the manuscript (in addition to the references M. Machida, et al., Int. J. Inorg. Mater., 2001. 3(6): p. 545-550, M.V. Ryzhkov, et al., Z. Phys. B - Condensed Matter, 1985. 59(1): p. 1-6, and K. Nishidate, et al., Mater. Res. Express, 2020. 7(6): p. 065505 for the electronic structure that were already in the original manuscript on page 8) to avoid any further confusion.

Comment 2:

The authors propose that La/Y doping leads to highly ordered structures, which contributes to a larger band gap (Eg) and enhanced photoelectrochemical performance, while Ce/Pb doping induces disorder, resulting in a smaller Eg and poorer performance. However, the causal relationship between structural ordering and band edge positions has not been clearly explained at a quantitative or theoretical level. The authors are advised to clarify whether any established physical models or literature support this correlation.

Response: We apologize for the confusion. The Ce- and Pb-doped samples show poor photoelectrochemical performance, which we attribute to limited electrochemical stability. This was already mentioned in the second paragraph on page 13, line 459 to 468. However, the Y doped sample also shows poor performance that can be attributed to limited electrochemical stability with mixed phases. The reviewer may have overlooked this. In general the Ce- and Pb-doping ions depopulate both the Bi3+ and Bi5+ sites, which enhances disorder in the structure. In addition, Pb-doping leads to a decrease in band gap, which is detrimental to the photoelectrochemical properties. We have now made corresponding clarifications on page 8 line 303 to 304 “which depopulates both the B1 and B2 sites in the structure, analogous to doping with M=Pb” and on page 12 line 439 to 441 However, an inferior PEC performance is anticipated for the M=Pb photoelectrode …,… and the rapid charge recombination.

Finally, Ce-doping leads to an increase in band-gap rather than a decrease. This is explicitly stated on page 8 line 302 to 306 “Doping with M=Ce……,…… for an enlarged Eg”, with a reference to the paper: R.J. Drost and W.T. Fu, Mater. Res. Bull., 1995. 30(4): p. 471-478.

Comment 3:

In the Ce, Pb, and Y-doped samples, the optical absorption spectra exhibit long tails extending to 1000 nm, but the Eg values estimated from the Tauc plots (<0.8 eV) are not adopted by the authors. It is suggested that the authors discuss the possible origins of these long tails (e.g., defect states, mid-gap states, sub-gap absorption) and clarify whether these Eg values represent actual band transitions or are related only to localized state transitions.

Response: We thank the reviewer for this comment, which is useful. It was already stated on page 7 line 261 to 267. We have now modified this sentence to include the suggestions from the reviewer: “The flat absorption tails in Figure 2(a) are probably caused by a considerable amount of defects relating to structural disorder, such as defect states, mid-gap states, sub-gap absorption, in the lattice of Ba2M0.4Bi1.6O6 when M=Ce or Pb, in line with similar absorption phenomena that were observed in previous investigations. The contribution of defects, e.g. lattice elemental vacancies or anti-sites in Ba2M0.4Bi1.6O6 when M=La or Pr to the UV-vis light absorption could also not be excluded for generating the in-gap states within the band gap of the materials”. We have also included 1) N. Zarrabi, et al., J. Phys. Chem. Lett., 2023. 14(13): p. 3174-3185, 2) Y. Wang, et al., eScience, 2024. 4(3): p. 100228 and 3) Y. Xu, Z. Wang and Y. Weng, J. Phys. Chem. C, 2024. 128(39): p. 16275-16290 additional references to provide further underpinning from literature data.

Comment 4:

Citing previous related studies would provide strong background support. The authors may consider referring to the following two papers, which have addressed related issues and provided valuable insights into perovskite systems: Nature Communications 7, 10636 (2016), Advanced Materials 32, 2003033 (2020).

Response: We thank reviewer’s suggestion. These two papers have now been referenced in the introduction of our manuscript, with the corresponding revisions made on page 1 line 32 to 33, Recent advances in layered perovskites (e.g. Bi2WO6) have also highlighted their promising photoelectrochemical performance”.

Reviewer 3 Report

Comments and Suggestions for Authors

I would recommend minor revision for this manuscript. The overall quality of the work is solid, and the findings are meaningful, especially given the growing interest in double perovskite systems beyond conventional halide structures. However, a few revisions are needed for improvement.

1. The manuscript does not provide a rationale for selecting this specific doping level. Why was the doping ratio M:Bi (0.4:1.6) chosen ? Please clarify whether this ratio was based on previous literature, empirical optimization, or structural considerations.

2. Why do the Ce and Pr doped samples exhibit much lower photocurrent, even though their light absorption profiles are comparable to that of the La-doped sample? The authors attribute the superior performance of the La-doped film to structural ordering and band alignment, but further clarification is needed to explain the poor performance of the other doped systems.

3. To simplify the electrochemical analysis section, Figure 8 and Figure 9 should be merged or reorganized. Consider combining them into a single figure or moving Figure 9 to Supporting Information to improve readability.

Author Response

#reviewer 3

Comments and Suggestions for Authors

I would recommend minor revision for this manuscript. The overall quality of the work is solid, and the findings are meaningful, especially given the growing interest in double perovskite systems beyond conventional halide structures. However, a few revisions are needed for improvement.
1. The manuscript does not provide a rationale for selecting this specific doping level. Why was the doping ratio M:Bi (0.4:1.6) chosen ? Please clarify whether this ratio was based on previous literature, empirical optimization, or structural considerations.

Response: We appreciate your comments. The choice is made based on our previous work (T. Guo, W.T. Fu and H.J.M. de Groot, J. Phys. Chem. C, 2024. 128(31): p. 13177-13189), where we have found optimal PEC performance for 0.4 concentration. This is now stated in the introduction on page 2 line 67, and the reference has been added.

2. Why do the Ce and Pr doped samples exhibit much lower photocurrent, even though their light absorption profiles are comparable to that of the La-doped sample? The authors attribute the superior performance of the La-doped film to structural ordering and band alignment, but further clarification is needed to explain the poor performance of the other doped systems.

Response: As has explained above in answers to the comments of reviewer 2, this can be attributed to the limited electrochemical stability as stated in the second paragraph on page 13, line 459 to 468. Corresponding revisions have also been made to the manuscript, please refer to page 7 line 261 to 267, page 8 line 303 to 305, and page 12 line 439 to 441.

3. To simplify the electrochemical analysis section, Figure 8 and Figure 9 should be merged or reorganized. Consider combining them into a single figure or moving Figure 9 to Supporting Information to improve readability.

Response: Thank you for your suggestion. Figure 9 has been moved to the supporting information and is now Figure S8.

Reviewer 4 Report

Comments and Suggestions for Authors

Journal: Nanomaterials

Manuscript number: nanomaterials-3646289

The paper presents a comprehensive study on the effects of various dopants (M = La, Ce, Pr, Pb, Y) on the structural, optical, electronic, and solar conversion performance of Ba2M0.4Bi1.6O6 double perovskites. Techniques such as UV-Vis spectroscopy, X-ray diffraction (XRD), and scanning electron microscopy (SEM) were employed to analyze the synthesized double perovskites.

The conclusions drawn are objective and well-supported. The subject matter is highly relevant to the Journal of Nanomaterials and is likely to engage the interest of its readers. The findings have potential applications in solar energy conversion.

In my opinion, the paper can be accepted after a minor revision.

Tthere are a few points that require clarification from the authors.

  1. P.3, line 113. The authors should provide the parameters of the Au substrate used to fabricate the samples with perovskite films, including thickness, surface smoothness, and initial dimensions.
  2. Figure 9, P.15. The authors should utilize the surface capacitance parameter instead of capacitance for the Mottky-Schottky curves to facilitate a comparison of the results with literature data.

Author Response

#reviewer 4

Comments and Suggestions for Authors

Journal: Nanomaterials

Manuscript number: nanomaterials-3646289

The paper presents a comprehensive study on the effects of various dopants (M = La, Ce, Pr, Pb, Y) on the structural, optical, electronic, and solar conversion performance of Ba2M0.4Bi1.6O6 double perovskites. Techniques such as UV-Vis spectroscopy, X-ray diffraction (XRD), and scanning electron microscopy (SEM) were employed to analyze the synthesized double perovskites. The conclusions drawn are objective and well-supported. The subject matter is highly relevant to the Journal of Nanomaterials and is likely to engage the interest of its readers. The findings have potential applications in solar energy conversion.

In my opinion, the paper can be accepted after a minor revision. There are a few points that require clarification from the authors.

1. P.3, line 113. The authors should provide the parameters of the Au substrate used to fabricate the samples with perovskite films, including thickness, surface smoothness, and initial dimensions.

Response: Thank you for your suggestion. Au substrate electrodes of 0.25 mm thickness and 25 x 25 mm initial dimensions were purchased (≥99.9975%, Thermo Scientific Chemicals). The electrodes were cut to dimensions of 1.25 x 2.5 cm2 and polished with sanding paper (Matador waterproof P1000, Starcke). These data have been added to the materials and methods on page 3 line 116 to 119.

2. Figure 9, P.15. The authors should utilize the surface capacitance parameter instead of capacitance for the Mott-Schottky curves to facilitate a comparison of the results with literature data.

Response: The Mott-Schottky analysis normalized to surface area in the former Figure 9 (now renumbered as Figure S8) uses the semiconductor capacitance Csc according to eq. 3, i.e. we apply the M-S equation with the electrolyte potential drop correction to the data. This follows the literature, in particular references 71) R. De Gryse, et al., J. Electrochem. Soc., 1975. 122(5): p. 711-712. and 73) F. Cardon and W.P. Gomes, J. Phys. D: Appl. Phys., 1978. 11(4): p. L63-L67, and including a recent literature ref.: 72) A. Hankin, et al., J. Mater. Chem. A, 2019. 7(45): p. 26162-26176 that is well cited. This is now further explained in the text following equation 3 on page 15, “The interfacial potential drop apparently originates from both the space charge region of the solid semiconductor and the electrical double layer in the liquid electrolyte”. In addition the Cdl at open circuit potential is provided in Table 3.

Round 2

Reviewer 2 Report

Comments and Suggestions for Authors

I think the revised manuscript addresses my concerns and can be accepted as it is.